# Tear Production, Intraocular Pressure, Ultrasound Biometric Features and Conjunctival Flora Identification in Clinically Normal Eyes of Two Italian Breeds of Chicken (*Gallus gallus domesticus*)

**DOI:** 10.3390/ani11102987

**Published:** 2021-10-17

**Authors:** Samanta Nardi, Federico Puccini Leoni, Viola Monticelli, Valentina Virginia Ebani, Fabrizio Bertelloni, Margherita Marzoni, Francesca Mancianti, Simonetta Citi, Giovanni Barsotti

**Affiliations:** 1Department of Veterinary Science, University of Pisa, Via Livornese lato Monte, 56122 Pisa, Italy; federico.puccini33@gmail.com (F.P.L.); simonetta.citi@unipi.it (S.C.); giovanni.barsotti@unipi.it (G.B.); 2Independent Researcher, St. Columb Major TR8 4JA, UK; violamon93@gmail.com; 3Department of Veterinary Sciences, University of Pisa, Viale delle Piagge 2, 56124 Pisa, Italy; valentina.virginia.ebani@unipi.it (V.V.E.); fabrizio.bertelloni@unipi.it (F.B.); margherita.marzoni@unipi.it (M.M.); francesca.mancianti@unipi.it (F.M.)

**Keywords:** tear production, intraocular pressure, ultrasound biometric features, conjunctival microflora, chicken, *Gallus gallus domesticus*

## Abstract

**Simple Summary:**

In Italy, chickens are used for egg production and as courtyard/domestic animals and consequently veterinarians need to know their general and specialist characteristics. One key area is normal ocular measurements in order to understand any pathological changes affecting the eyes. For an accurate diagnosis and better management of ophthalmic diseases in chickens, this paper describes the normal values for the evaluation of ocular tear production, intraocular pressure, and biometric measurements of the eyes and on the microbial and cultural flora normally present in the conjunctival sac in two Italian chicken breeds.

**Abstract:**

Given the abundance of chickens in Italy, it is important for veterinarians to know the normal state of chickens’ eyes in order to identify any ophthalmic pathological changes. The aim of this study was to determine the normal values of select ocular parameters and to evaluate conjunctival microflora in two Italian chicken breeds. Sixty-six healthy chickens underwent a complete ophthalmic examination, which included a phenol red thread test (PRTT) for the evaluation of tear production and the assessment of intraocular pressure by rebound tonometry. B-mode ultrasound biometric measurements and conjunctival microflora identification were also performed in twenty-seven chickens. Mean PRTT was 23.77 ± 2.99 mm/15 s in the Livorno breed and 19.95 ± 2.81 mm/15 s in the Siciliana breed. Mean intraocular pressure was 14.3 ± 1.17 mmHg in the Livorno breed and 14.06 ± 1.15 mmHg in the Siciliana breed. Reference ranges for morphometric parameters were reported in the two breeds. Twenty-three chickens (85.18%) were bacteriologically positive. Chlamydia spp. antigen was detected in 14.81% of chickens. No positive cultures were obtained for fungi. Normal reference range values for selected ophthalmic parameters were obtained in clinically healthy chickens, which could facilitate accurate diagnosis and better management of ophthalmic diseases in these animals.

## 1. Introduction

In Italy there are various breeds of chicken (*Gallus gallus domesticus*—Linnaeus, 1758) and in addition to intensive chicken farms, it is very common to own a small number of chickens both for egg production and as courtyard/domestic animals. For this reason, chickens can require the intervention of a veterinarian for health problems, including ophthalmic diseases. There is thus an increasing need for eye examinations in chickens.

Ophthalmologic examination is routine practice in veterinary ophthalmology also for avian species. This examination in birds is essential to assess the status of the eyes which can also be considered as an indicator of overall health status of the animal. However, the ocular anatomy of birds varies among different species, and it is thus impossible to use the same ocular parameters and measurements for all species [1,2,3,4,5,6,7,8,9,10,11,12,13,14,15,16,17,18,19,20,21,22,23,24,25,26].

Although the anatomy of the chicken eye has been well described [27,28], few articles report the normal values of some ocular parameters such as tear production, intraocular pressure, and ultrasound biometric measurements [1,3,7,17,29]. In addition, to the best of our knowledge, no data on conjunctival flora have been reported in this species. Knowledge of the conjunctival flora in birds is important when a corneal injury occurs in order to increase the accuracy of treatments. In fact, corneal ulcers can become secondarily infected by opportunistic flora of the conjunctival sac [22,30,31,32].

The aim of this study was thus to describe, in normal eyes of healthy chickens, various ophthalmic parameters including tear production, intraocular pressure, ultrasound biometric features, and the normal bacterial and fungal flora present in the conjunctival sac.

## 2. Materials and Methods

Animals were reared according to the principles stated in Directive 2010/63/EU. Sixty-six chickens (31 males, 35 females) without signs of ocular or systemic diseases were used. Approval for this study was obtained from the Ethics Committee on Animal Experimentation of the University of Pisa D. lgs.vo 26/2014 prot. n 17/2018.

### 2.1. Rearing Conditions of Chickens

Chickens used for the trial came from two Italian breeds, the Siciliana (S) and the Livorno and two plumage varieties, white (WL) and black (BL). These breeds are local slow-growing genotypes and are the most common autochthonous breeds for egg production and among the best breeds for free-range systems due to their rusticity. The birds were hatched at the poultry farm of the Veterinary Sciences Department of Pisa University. During the study, chickens were reared in partially roofed outdoor pens (3 m × 9 m, 3 m height; 3 m^2^/bird) on sandy ground and equipped with perches for open-air rearing throughout the growing and reproductive periods and for the rest of their lifespan.

The rearing environmental parameters (temperature, relative humidity levels, photoperiod) changed according to the season.

### 2.2. Data Collection

A total of 31 (15 males, 16 females) Siciliana and 35 (16 males, 19 females) Livorno chickens, aged 276 to 375 days (39 to 53 weeks), had their eyes tested. Tear production and intraocular pressure were always tested in the morning (from 10:00 to 12:00). The live body weight of the 66 breeders was determined at the moment of ophthalmologic examination. The study was performed in March and the mean humidity was 71% and temperature 15 °C. Ultrasound biometric measurements and collection of samples for conjunctival flora identification were performed in 27 out of 66 chickens because the birds that exhibited excessive stress during the procedures were excluded.

### 2.3. Ophthalmologic Examination

All the animals were considered free from ophthalmic abnormalities, as determined by an ophthalmologic examination performed before data collection. The eye and periocular region were examined in ambient light for gross abnormalities. Menace response and palpebral and corneal reflex tests were also performed, taking into account the difficulties in assessing the response in chickens.

Tear production was assessed in each eye using a commercial phenol red thread test (PRTT) kit (PRT-TEST; Tianjin Jingming New Technological Development Co. Ltd., London, UK). The end with the 3 mm angle for the test was placed inside the lower fornix at a point approximately one third of the eyelid width from the temporal cantus. Tear production was recorded in millimeters, wetting after 15 s.

The adnexa and anterior segment of both eyes were examined with a portable slit-lamp biomicroscope (Kowa SL-17^®^, Kowa Company, Tokyo, Japan). This procedure as well as the assessment of pupillary light reflexes were carried out in a dark room. Intraocular pressure (IOP) was measured using rebound tonometry (TonoVet^®^, iCare, Vantaa, Finland) with the P setting, which is not specific to any species. The P setting was chosen due to the lack of a specific tonometer internal calibration table for measuring IOP in birds. During each IOP assessment, birds were gently restrained, and any pressure on the neck, globe, and periocular region was avoided.

The first three IOP measurements with low/no error were recorded for each eye, each measurement being the average that the rebound tonometer automatically performed on six valid bounces against the cornea. The average of the three measurements was considered the IOP value per eye. The IOP was taken starting each time with the left eye. The ocular fundus was observed using a binocular indirect ophthalmoscope (Omega 180; Heine, Berlin, Germany) with a 20 or 30 D lens, without pharmacological pupil dilation. The fundus examination was performed in a dark room without the use of a mydriatic agent, only minimizing the intensity of the examination light. Fluorescein staining was also performed after completion of fundoscopy. All the measurements were performed by the same operator (GB).

Ocular ultrasonography was performed in twenty-seven chickens and specimens were collected for cultural evaluation of the flora of the conjunctival sac.

### 2.4. Ultrasound Biometric Measurements

In twenty-seven chickens (54 eyes), transpalpebral B-mode ultrasonographic examinations were performed using an ultrasound system, a high frequency linear probe (12 MHz) and abundant ultrasound gel. The chickens were manually restrained. No general and/or topical anesthesia was required for transpalpebral ultrasonographic examinations since the eyelids were closed, and the cornea was not exposed. Axial and transvers scans of the chicken globe were performed. The left eye was assessed before the right eye.

The following morphometric parameters were measured: the distance between the cornea and the anterior capsule of the lens (D1), the distance between the posterior capsule of the lens and the optical papilla (D2), the thickness of the lens (D3), and the axial (D4) and transverse (D5) lengths of the globe. All ultrasound examinations were performed by the same experienced radiologist (SC) in order to reduce interobserver variability.

### 2.5. Conjunctival Bacterial and Fungal Flora Identification

In twenty-seven chickens (54 eyes), materials for bacterial and fungal culture were collected at the inferior conjunctival fornix. The chickens were restrained manually, and three swabs (TRANSYSTEM AMIES Agar GelWith Charcoal, Biolife Italiana Srl, Milan, Italy) were taken from both eyes. One swab was for bacterial culture, one for Chlamydia spp. antigen identification, and the third swab for fungal flora culture was executed one week after material collection for bacterial culture. All the samples were collected by the same operator (GB), without topical anesthesia by rolling the sterile swab on the conjunctival surface, thus avoiding contact with the eyelid margin and cornea.

Enumeration of mesophilic aerobic and heterotrophic anaerobe microorganism (total bacterial count—TBC) was determined for each eye: one swab was mixed with 9 mL of sterile saline solution (1:10 dilution), vortexed and processed. Ten-fold serial dilutions (1:100–1:10,000) were performed, and 1 mL from each of them was inoculated in a sterile Petri dish. Next, 15 mL of Tryptic Glucose yeast Agar (PCA) (Biolife Italiana Srl, Milan, Italy) cooled to 45 °C was added to each plate. Inoculum and medium were carefully mixed, left to solidify, and another 4 mL of APC was added.

After incubation at 30 °C for 72 h, colonies on plates containing more than 10 and less than 300 colonies were counted. TBC was expressed as colony forming units for mL (CFU/mL). In addition, the same swab was inoculated onto a blood-agar plate, incubated at 37 °C, and examined for bacterial growth after 24, 48, and 72 h. Developed colonies were examined by Gram staining, and submitted to typing employing specific media and biochemical tests.

One swab for each eye was employed to detect the presence of *Chlamydia* spp. antigen using the commercial immunoenzymatic test Clearview Chlamydia MF (Inverness Medical, Waltham, MA, USA).

Samples for mycological culture were seeded onto malt extract agar (MEA, Oxoid, Milan, Italy), and gentamicin was added to avoid bacterial contamination. The samples were then incubated at 25 °C and examined daily from day 4 post-incubation, over a 10-day period to identify mycotic growth.

### 2.6. Statistical Analysis

For each parameter analyzed, the average values +/− the standard deviation, the median, the maximum and the minimum values were reported. The D’Agostino and Pearson test was used to investigate the Gaussian distribution of quantitative variables. The differences relating to D1, D2, D3, D4, D5, PRTT, and IOP between OD (oculus dexter) and OS (oculus sinister) and between the various breeds were investigated by Student’s t test for parametric data or by the Mann–Whitney test for nonparametric data.

The mean PRTTs, IOPs, D1s, D2s, D3s, D4s, and D5s of the right eye and left eye were considered as a single value for each bird (single value per animal), because there were no differences between the two eyes. A possible correlation between the weight of every chicken and each biometric parameter was evaluated by the Pearson correlation coefficient for parametric data or by the Spearman correlation coefficient for nonparametric data. In the same way, a possible correlation between the weight and the age of each chicken and PRTT and IOP values was investigated. *p* values < 0.05 were considered significant.

## 3. Results

### 3.1. Ophthalmologic Examination

The mean PRTT of all the chickens was 21.98 ± 3.46 mm/15 s, with 23.77 ± 2.99 mm/15 s in the Livorno breed, and 19.95 ± 2.81 mm/15 s in the Siciliana breed. There were no significant differences in PRTT between OD and OS in the Livorno (*p* = 0.2787) and the Siciliana (*p* = 0.4448) breeds, but there was a significant difference between the two breeds themselves (*p* < 0.0001). Descriptive and inferential statistics of PRTT data are presented in Table 1a–c.

The mean IOP of all the chickens was 14.19 ± 1.16 mmHg, with 14.3 ± 1.17 mmHg in the Livorno breed and 14.06 ± 1.15 mmHg in the Siciliana breed. There were no significant differences in IOP between OD and OS in the Livorno (*p* = 0.1157) and the Siciliana (*p* = 0.3197) breeds, and no differences between the two breeds themselves (*p* = 0.4145). Descriptive and inferential statistics of IOP data are presented in Table 2a–c.

No correlations between the weight and PRTT (*p* = 0.1419) or the weight and IOP (*p* = 0.6865) were detected. No correlations between the age and IOP (*p* = 0.6182) were observed in either breed. However, there was a direct correlation between the age of chickens and PRTT (*p* = 0.00079) in both breeds.

### 3.2. Ultrasound Biometric Measurements

When an ultrasound is performed in B-mode, the cornea is hyperechoic and convex, and the anterior chamber is anechoic. The lens appears anechoic and ovoid and is delimited by two curvilinear hyperechoic lines corresponding to the anterior and posterior lens capsule. The vitreous chamber is anechoic with the pecten visible as a hyperechoic, millimetric, tubular-shaped structure, projecting into the vitreous chamber from the retina. The retina, the choroid, and the sclera are not ultrasonographically distinguishable since they appear as a single concave hyperechoic line at the posterior limit of the globe. The distance between the cornea and the anterior lens capsule (D1) (Figure 1A), the distance between the posterior lens capsule and the optical papilla (D2) (Figure 1B), the lens thickness (D3) (Figure 2B), and the axial (D4) (Figure 2A) and transverse (D5) (Figure 3) lengths of the globe for each breed and for all breed pools are shown in Table 3a–c.

No differences were observed in the ocular ultrasound appearance, both within the same breed and between the two breeds. There was a significant difference for D1 between the Livorno and the Siciliana breeds (*p* = 0.0301), while no significant differences were observed for D2 (*p* = 0.9800), D3 (*p* = 0.1087), D4 (*p* = 0.0877), and D5 (*p* = 0.7212) between the Livorno and the Siciliana breeds. There was a significant direct correlation between D1 and the weight of the chickens (*p* = 0.0329), while D2 (*p* = 0.7663), D3 (*p* = 0.3923), D4 (*p* = 0.3715), and D5 (*p* = 0.8891) were independent of weight.

### 3.3. Conjunctival Bacterial and Fungal Flora Identification

Among the 27 chickens examined, 23 (85.18%) were bacteriologically positive for one or both eyes. Twelve (44.44%) chickens had a positive culture (including presence of *Chlamydia* spp.) in both eyes, and of these twelve chickens, ten (37.03%) had the same bacterial genus in both eyes.

The CFU/mL values ranged from <100 to >10^5^. Bacteria belonging to the following genera were cultured: *Corynebacterium* spp. was isolated from 16 (29.62%) eyes, *Bacillus* spp. from 14 (25.92%), *Pasteurella* spp. from 5 (9.25%), *Staphylococcus* spp. coagulase negative from 5 (9.25%), *Staphylococcus* spp. coagulase positive from 4 (7.40%), *Ochrobacterium* spp. from 2 (3.70%), and *Enterobacter* spp. from 1 (1.85%).

*Chlamydia* spp. antigen was detected in both eyes of four (14.81%) chickens. The results obtained for each eye are reported in Table 4.

No positive cultures were obtained for fungi.

## 4. Discussion

Although there are many studies regarding the normal values of tear production and intraocular pressure (IOP) in many bird species, in chickens there are few studies with differing results [1,7,29,33].

In the present study, we used a commercially available phenol red thread test (PRTT) kit to determine tear production in chickens for two reasons: firstly, because the palpebral fissure is small and PRTT is easier to use than other tests; secondly, it takes less time (15 s) to get the results compared with the Schirmer Tear Test (1 min).

Our results indicate that the mean normal tear production in adult chickens with PRTT is 21.98 ± 3.46 mm/15 s: 23.77 ± 2.99 mm/15 s in the Livorno breed and 19.95 ± 2.81 mm/15 s in the Siciliana breed, with a significant difference between the two breeds (*p* < 0.0001).

Our mean results in the Livorno breed were similar to those of diurnal raptors (23.5 ± 8.9 mm/15 s) [34], while our mean results in the Siciliana breed were similar to those of large Psittaciformes (OD: 19.8 ± 4.3 mm/15 s; OS: 20.1 ± 3.9 mm/15 s) [35] and to those of the common mynah (19.2 ± 2.5 mm/15 s) [36].

Given that there is an inter-specific difference in normal tear production in birds, reference values for each species need to be established (Table 5) [8,9,10,18,19,20,21,22,23,25,26,29,34,35,36,37,38,39,40,41,42,43,44,45,46,47].

Although normal tear production in birds can be evaluated with several methods, such as PRTT and STTI, they are not equivalent and reference values should be established for each method [8,9,22,29,34,38,46].

There is also an intraspecific difference among chicken breeds. In fact, we found significant differences between the Livorno and the Siciliana breeds. Tear production in the two breeds was evaluated at the same time and in the same environmental conditions. This indicates that tear production differences were only due to breed.

Although Table 5 highlights that there are many articles reporting tear production values in birds, we can only partially compare our results with those by Fornazari et al. (2018) who performed PRTT in 42-day-old chicks and showed values of 25.58 ± 4.8 mm/15 s. The difference in tear production could be due to different breeds of chicken tested and to environmental factors, such as humidity, temperature, wind, dust, and ammonia levels. Our population and Fornazari’s were born and reared in different environments, with different temperatures and humidity. However, we do not know the exact levels of dust and ammonia. In fact, a limitation of our study was the lack of any ammonia level measurements. However, because we used an open-air rearing system, it is likely that the ammonia levels were low and that they did not influence tear production.

Our study revealed a direct correlation between the age of the chickens and PRTT (*p* = 0.00079) in both breeds. This can be explained by different stages of maturation of the lacrimal gland while the chicks are developing, but also by the limited corneal reflex in young animals, as already reported in humans and in other animals [48,49,50,51]. Fornazari et al. (2018) examined two populations of two differently aged chickens (5 and 42-day-old chicks), and they also reported that tear production increases with age [29].

There are various tonometers for determining the intraocular pressure in animal patients. In chickens we used a rebound tonometer because it is easier to use when the palpebral fissure is small, and it does not require a topical anesthesia. All IOP measurements were obtained using the P setting on the rebound tonometer, which is not specific to any species, because there is no internal calibration table for tonometers for measuring IOP in birds.

Many IOP values have been reported using different tonometers in many bird species (Table 6) [1,4,6,7,8,9,10,11,15,16,18,19,20,22,23,24,25,26,33,34,38,39,41,42,43,44,45,46,47,52].

This means that it is not possible to compare the results from different devices and between different species. Instead, comparisons are possible when the same type of tonometer is used, with the same calibration, and in the same bird species. We can thus only compare our mean results (14.19 ± 1.16 mmHg) with those of Prasher et al. (2007). They reported three different results (expressed as mean value ± standard error) for three different chicken lines: 16.3 ± 0.2 mmHg for an egg-layer line, 16.1 ± 0.2 mmHg for a broiler line, and 17.51 ± 0.13 mmHg for an advanced layer–broiler intercross line.

Our results and Prasher’s (2007) are similar, but do not overlap. This may depend on the different ages and weights between our chicken population and Prasher’s. In fact, our animals were more than 39-weeks-old (39 to 53 weeks) and can be considered as young adults/adults, while Prasher’s were only 3 weeks old.

Another factor to assess is the time at which the IOP was measured in the subjects examined. In fact, the IOP in normal eyes of chickens is high during the day, and low in the middle of the night [1]. Therefore, the variation between our results and Prasher’s could also be due to the different time at which the IOP measurements were performed: from 10:00 to 12:00 a.m. and from 12:00 to 16:00 p.m., respectively.

Our results provide reference values for IOP in young-adult/adult subjects belonging to two different breeds of the same bird species. Our results showed that there was no significant difference between IOP and the two breeds of chickens, and between IOP and age (39 to 53 weeks) in our young-adult/adult population. In fact, Table 6 highlights a strong difference in IOP values between different species of birds when IOP was also assessed using the same device.

Few studies have investigated ocular dimensions in chicken, although one study evaluated the size of the various ocular structures in two-week-old chickens, without breed distinction and using A-mode and Optical Low-Coherence Interferometry [17]. The ultrasonographic appearance of the eyes of chickens in our study is the same as that described for other avian species [12,13]. We observed no significant morphometric differences between OD and OS. This is in agreement with findings for other species both within each breed and within an all breed pools [12,17,53].

The ultrasonographic appearance of the chickens’ eyes in our study was the same as that described for other avian species [12,13] and the average distance between the cornea and the anterior capsule of the lens was 1.68 mm for OS and 1.66 for OD, a typical measurement of non-passerines in general [13]. We observed no significant morphometric differences between OD and OS, in agreement with other findings for other species both within each breed and within all breed pools [12,17,53]. The lack of statistical morphometric differences between the Livorno and Siciliana breeds could indicate that these two chicken breeds have similar ocular dimensions, except for D1. The present study indeed showed that there is a significant difference for D1 between the Livorno and the Siciliana breeds, and a correlation was found between D1 and the weight of the chickens. However, an ultrasound of the anterior chamber may be complicated to assess due to its compressibility and proximity to the probe, and for this reason the measurement of D1 may be distorted [54]. The correlation between D1 and the weight of chickens, as well as the difference in D1 between the two breeds, might thus be insignificant. That being said, we can affirm that the B-mode ultrasound is a useful, simple, and safe tool for studying eyes in chickens, since it gives information on the size and appearance of the globe and its internal structures without the need for anesthesia. Biometric measurements of the eye can be used to detect changes in the shape or size of the eye, as occurs in glaucoma [13]. In addition, the average reference measurement of the intraocular structures in healthy chickens could be useful in avian clinical ophthalmology and when chickens are used as experimental models in human ophthalmology. Chickens are commonly used as an animal model for the study of myopia, ocular albinism, retinal dystrophies, coloboma, glaucoma, keratoconus, retinal detachment, retinal degeneration, and ocular tumors [27,55,56].

To the best of our knowledge, there are currently no data on conjunctival bacterial flora in poultry. Previous studies have been limited to the bacteria present in the eyes of wild birds. Our results are in reasonable agreement with those found by Dupont et al. (1994), who studied the bacterial and fungal flora in the healthy eyes of Falconiform and Strigiform raptors from Canada. They isolated bacteria of the genera *Bacillus, Corynebacterium, Staphylococcus, Pasteurella*, and *Enterobacter* with prevalences, in some cases, similar to those detected in our investigation. In particular, Gram-positive bacteria were the microorganisms most frequently isolated in both studies, although Dupont et al. found *Staphylococcus* spp. coagulase negative with the highest prevalence (49.5%).

The detection of these bacterial genera in healthy eyes is not surprising, considering that they are usually non-pathogenic. However, in some circumstances, such as injuries or inflammations, they can cause ocular infections. Some bacteria in healthy eyes seem to play an important role in their defense mechanisms, taking nutrients from invading organisms or secreting substances with antimicrobial properties [27,30,57].

On the other hand, in our study the detection of *Chlamydia* spp. in the examined eyes suggests the circulation of this pathogen in the environment where the chickens lived. *Chlamydia psittaci* frequently infects domestic and wild birds without inducing disease. Birds contract chlamydias through oral and/or inhalation, but also through conjunctival contamination by dust. Infected birds can develop conjunctivitis and respiratory and enteric clinical signs, mainly when the birds are stressed [58].

In the present study, no positive fungal cultures were achieved. Our data cannot be compared with similar studies from the literature, since to the best of our knowledge, no papers have investigated fungal flora in chickens. There appear to be only two studies dealing with the occurrence of fungi in birds’ eyes and both were carried out on birds of prey. In those studies, *Cladosporium* spp. and *Aspergillus* spp. were cultured from one and 2 eyes, respectively, out of 97 subjects [30], while *Aspergillus fumigatus* and *Candida* spp. were obtained from 1 and 2 birds, respectively, out of 51 raptors [34]. The occurrence of fungi in the conjunctivae is believed to be transitory and resulting from environmental exposure. These features could explain their low prevalence [59] considering that the chickens in our study live outdoors and probably there are no environmental conditions for the stabilization of the fungi in the conjunctival sac. Fungi have been more frequently isolated from horses [60,61,62,63,64], donkeys [65,66], cattle [67], dogs [68], rabbits [69], turtles, and tortoises [70] and the genera and species recovered seem to reflect a transient seeding from the environment.

## 5. Conclusions

In conclusion, our study performed on clinically healthy chickens reported normal reference range values for selected ophthalmic parameters, including ultrasound biometric measurements. These reference ranges can facilitate accurate diagnosis and better management of ocular diseases, preventing diagnostic misinterpretation during the ophthalmic examination. In healthy chickens we also found evidenced of conjunctival microflora composed of bacteria. Knowledge of the conjunctival flora in chickens is important to increase the accuracy of treatments in the case of corneal injuries. In fact, corneal ulcers can often become secondarily infected by opportunistic flora of the conjunctival sac.

## Figures and Tables

**Figure 1 animals-11-02987-f001:**
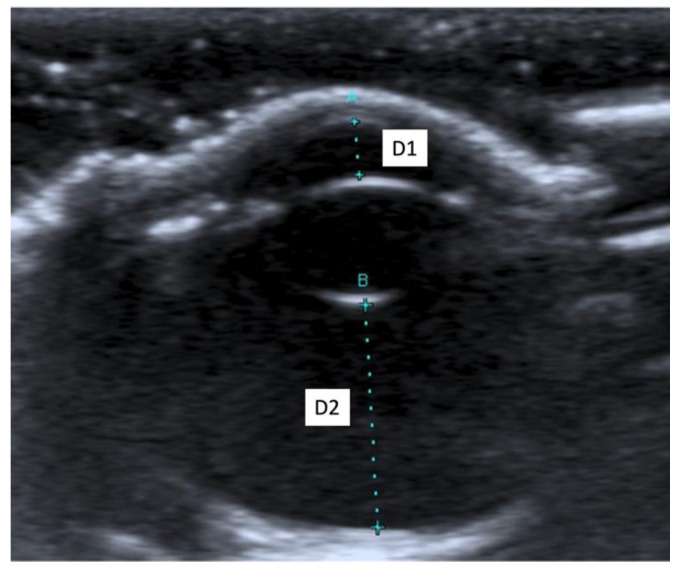
(**A**) The distance between cornea and anterior lens capsule (D1); (**B**) the distance between posterior lens capsule and optic papilla (D2).

**Figure 2 animals-11-02987-f002:**
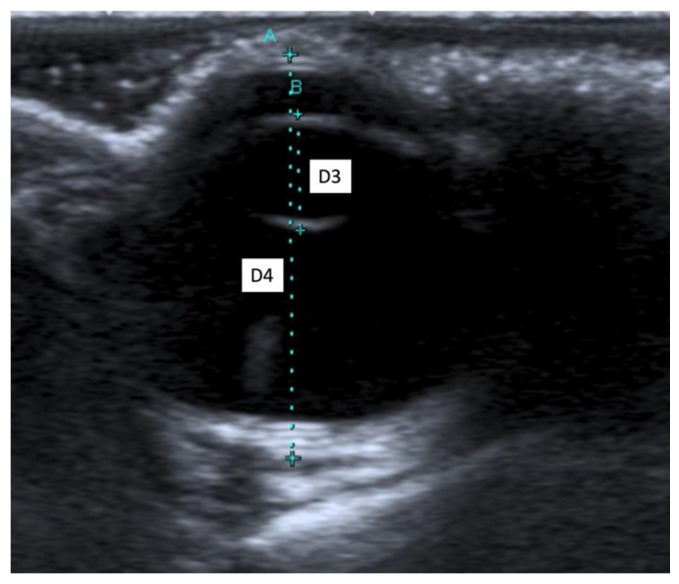
(**A**) The axial length of the globe (D4); (**B**) the lens thickness (D3).

**Figure 3 animals-11-02987-f003:**
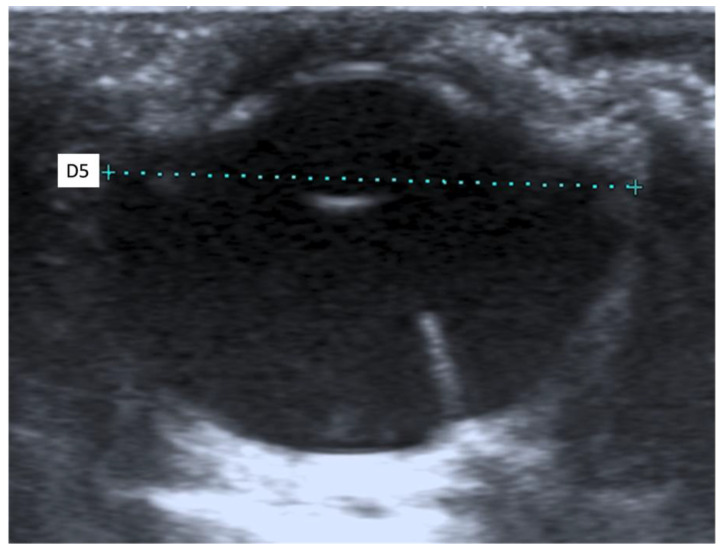
The transverse length of the globe (D5).

**Table 1 animals-11-02987-t001:** (**a**) Descriptive and inferential statistics of PRTT data in both breeds; (**b**) descriptive and inferential statistics of PRTT data in the Livorno breed; (**c**) descriptive and inferential statistics of PRTT data in the Siciliana breed.

(**a**)				
Both breeds	OS ^1^ (*n* ^3^ = 66)	OD ^2^ (*n* ^3^ = 66)	OS ^1^ + OD ^2^ (*n* ^3^ = 132)	Mean OS ^1^ + Mean OD ^2^
Mean (mm/15 s ^5^)	22.08	21.88	21.98	21.98
SD ^4^ (mm/15 s ^5^)	4.65	3.83	4.24	3.46
Median (mm/15 s ^5^)	22.00	21.00	22.00	22.00
Maximum (mm/15 s ^5^)	30.00	30.00	30.00	30.00
Minimum (mm/15 s ^5^)	12.00	15.00	12.00	16.00
(**b**)				
The Livorno breed	OS ^1^ (*n* ^3^ = 35)	OD ^2^ (*n* ^3^ = 35)	OS ^1^ + OD ^2^ (*n* ^3^ = 70)	Mean OS ^1^ + Mean OD ^2^
Mean (mm/15 s ^5^)	24.26	23.29	23.77	23.77
SD^4^ (mm/15 s ^5^)	3.85	3.59	3.73	2.99
Median (mm/15 s ^5^)	25.00	24.00	24.00	24.00
Maximum (mm/15 s ^5^)	30.00	30.00	30.00	30.00
Minimum (mm/15 s ^5^)	15.00	17.00	15.00	17.50
(**c**)				
The Siciliana breed	OS ^1^ (*n* ^3^ = 31)	OD ^2^ (*n* ^3^ = 31)	OS ^1^ + OD ^2^ (*n* ^3^ = 62)	Mean OS ^1^ + Mean OD ^2^
Mean (mm/15 s ^5^)	19.61	20.29	19.95	19.95
SD ^4^ (mm/15 s ^5^)	4.27	3.50	3.89	2.81
Median (mm/15 s ^5^)	19.00	20.00	19.50	20.00
Maximum (mm/15 s ^5^)	29.00	26.00	29.00	25.00
Minimum (mm/15 s ^5^)	12.00	15.00	12.00	16.00

^1^ oculus sinister; ^2^ oculus dexter; ^3^ number of eyes; ^4^ standard deviation; ^5^ seconds.

**Table 2 animals-11-02987-t002:** (**a**) Descriptive and inferential statistics of IOP data in both breeds; (**b**) descriptive and inferential statistics of IOP data in the Livorno breed; (**c**) descriptive and inferential statistics of IOP data in the Siciliana breed.

(**a**)				
Both breeds	OS ^1^ (*n* ^3^ = 66)	OD ^2^ (*n* ^3^ = 66)	OS ^1^ + OD ^2^ (*n* ^3^ = 132)	Mean OS ^1^ + Mean OD ^2^
Mean (mmHg)	14.24	14.14	14.19	14.19
SD ^4^ (mmHg)	1.30	1.26	1.28	1.16
Median (mmHg)	14.00	14.00	14.00	14.00
Maximum (mmHg)	18.00	17.00	18.00	17.50
Minimum (mmHg)	11.00	11.00	11.00	11.00
(**b**)				
The Livorno breed	OS ^1^ (*n* ^3^ = 35)	OD ^2^ (*n* ^3^ = 35)	OS ^1^ + OD ^2^ (*n* ^3^ = 70)	Mean OS ^1^ + Mean OD ^2^
Mean (mmHg)	14.54	14.06	14.30	14.30
SD ^4^ (mmHg)	1.27	1.28	1.29	1.17
Median (mmHg)	14.00	14.00	14.00	14.00
Maximum (mmHg)	18.00	17.00	18.00	17.50
Minimum (mmHg)	11.00	12.00	11.00	11.50
(**c**)				
The Siciliana breed	OS ^1^ (*n* ^3^ = 31)	OD ^2^ (*n* ^3^ = 31)	OS ^1^ + OD ^2^ (*n* ^3^ = 62)	Mean OS ^1^ + Mean OD ^2^
Mean (mmHg)	13.90	14.23	14.06	14.06
SD ^4^ (mmHg)	1.27	1.26	1.27	1.15
Median (mmHg)	14.00	14.00	14.00	14.00
Maximum (mmHg)	16.00	17.00	17.00	16.00
Minimum (mmHg)	11.00	11.00	11.00	11.00

^1^ oculus sinister; ^2^ oculus dexter; ^3^ number of eyes; ^4^ standard deviation.

**Table 3 animals-11-02987-t003:** (**a**) Morphometric results in both breeds; (**b**) morphometric results in the Livorno breed; (**c**) morphometric results in the Siciliana breed.

(**a**)															
Both breeds	OS ^1^ (*n* ^3^ = 27)					OD ^2^ (*n* ^3^ = 27)					Mean OS ^1^ + Mean OD ^2^				
	D1 ^5^	D2 ^6^	D3 ^7^	D4 ^8^	D5 ^9^	D1 ^5^	D2 ^6^	D3 ^7^	D4 ^8^	D5 ^9^	D1 ^5^	D2 ^6^	D3 ^7^	D4 ^8^	D5 ^9^
Mean (mm)	1.68	7.41	4.29	15.09	18.01	1.66	7.48	4.32	15.16	18.07	1.67	7.44	4.30	15.12	18.04
SD ^4^ (mm)	0.16	0.18	0.11	0.44	0.49	0.16	0.20	0.17	0.37	0.49	0.12	0.17	0.12	0.37	0.43
Median (mm)	1.70	7.40	4.30	15.20	18.10	1.70	7.50	4.30	15.20	18.00	1.65	7.45	4.30	15.05	18.05
Maximum (mm)	2.10	7.70	4.50	15.80	18.80	1.90	7.80	4.70	15.90	18.90	1.95	7.75	4.55	15.80	18.85
Minimum (mm)	1.40	7.00	4.10	14.30	16.80	1.30	7.10	4.10	14.60	17.30	1.45	7.05	4.10	14.50	17.25
(**b**)															
The Livorno breed	OS ^1^ (*n* ^3^ = 15)					OD ^2^ (*n* ^3^ = 15)					Mean OS ^1^ + Mean OD ^2^				
	D1 ^5^	D2 ^6^	D3 ^7^	D4 ^8^	D5 ^9^	D1 ^5^	D2 ^6^	D3 ^7^	D4 ^8^	D5 ^9^	D1 ^5^	D2 ^6^	D3 ^7^	D4 ^8^	D5 ^9^
Mean (mm)	1.70	7.41	4.26	14.95	18.01	1.73	7.48	4.28	15.08	18.02	1.72	7.44	4.27	15.02	18.01
SD ^4^ (mm)	0.20	0.17	0.13	0.42	0.50	0.12	0.18	0.16	0.38	0.45	0.12	0.15	0.13	0.36	0.39
Median (mm)	1.70	7.40	4.30	14.90	18.10	1.70	7.50	4.20	15.00	18.00	1.70	7.45	4.25	15.00	18.05
Maximum (mm)	2.10	7.70	4.50	15.70	18.70	1.90	7.80	4.60	15.90	18.90	1.95	7.75	4.55	15.80	18.80
Minimum (mm)	1.40	7.20	4.10	14.50	16.80	1.50	7.10	4.10	14.60	17.30	1.55	7.15	4.10	14.55	17.25
(**c**)															
The Siciliana breed	OS ^1^ (*n* ^3^ = 15)					OD ^2^ (*n* ^3^ = 15)					Mean OS ^1^ + Mean OD ^2^				
	D1 ^5^	D2 ^6^	D3 ^7^	D4 ^8^	D5 ^9^	D1 ^5^	D2 ^6^	D3 ^7^	D4 ^8^	D5 ^9^	D1 ^5^	D2 ^6^	D3 ^7^	D4 ^8^	D5 ^9^
Mean (mm)	1.66	7.41	4.32	15.30	18.03	1.57	7.47	4.37	15.25	18.13	1.62	7.44	4.35	15.26	18.08
SD ^4^ (mm)	0.10	0.21	0.08	0.40	0.51	0.15	0.23	0.16	0.35	0.55	0.10	0.19	0.09	0.33	0.50
Median (mm)	1.70	7.40	4.30	15.35	18.00	1.60	7.50	4.35	15.35	18.05	1.62	7.45	4.35	15.43	18.03
Maximum (mm)	1.80	7.70	4.40	15.80	18.80	1.80	7.80	4.70	15.70	18.90	1.75	7.70	4.55	15.60	18.85
Minimum (mm)	1.50	7.00	4.20	14.30	17.10	1.30	7.10	4.20	14.70	17.30	1.45	7.05	4.20	14.50	17.45

^1^ oculus sinister; ^2^ oculus dexter; ^3^ number of eyes; ^4^ standard deviation; ^5^ distance between cornea and anterior lens capsule; ^6^ distance between posterior lens capsule and optic papilla; ^7^ lens thickness; ^8^ axial length of the globe; ^9^ transverse length of the globe.

**Table 4 animals-11-02987-t004:** Conjunctival bacterial flora results.

Animals	OD ^a^		OS ^b^		Chlamydia
	Bacteria	TBC ^c^ (CFU/mL)	Bacteria	TBC ^c^ (CFU/mL)	
1	*Bacillus* spp.	2.2 × 10^2^	Negative	9.2 × 10^2^	Negative
2	Negative	3.2 × 10^3^	*Bacillus* spp./*Staphylococcus* spp. coagulase-negative	2.5 × 10^2^	Negative
3	*Corynebacterium* spp.	1.9 × 10^2^	Negative	5 × 10	Negative
4	Negative	2.3 × 10^2^	Negative	5.1 × 10^2^	Positive
5	Negative	3.9 × 10^2^	Negative	7 × 10	Negative
6	*Corynebacterium* spp./*Staphylococcus* spp. coagulase-positive	1.17 × 10^4^	*Corynebacterium* spp.	7.8 × 10^2^	Negative
7	*Bacillus* spp.	5.6 × 10^3^	*Bacillus* spp./*Staphylococcus* spp. coagulase-negative	1.06 × 10^4^	Negative
8	*Bacillus* spp.	6.9 × 10^3^	*Bacillus* spp.	2.3 × 10^2^	Negative
9	Negative	5.2 × 10^2^	*Corynebacterium* spp.	1.12 × 10^5^	Positive
10	Negative	3.3 × 10^2^	Negative	4.1 × 10^2^	Negative
11	Negative	3.1 × 10^3^	*Bacillus* spp./*Staphylococcus* spp. coagulase-negative	2.11 × 10^4^	Negative
12	*Corynebacterium* spp./*Staphylococcus* spp. coagulase-positive	1.31 × 10^4^	*Corynebacterium* spp.	8.2 × 10^3^	Negative
13	*Pasteurella* spp.	4.3 × 10^3^	*Staphylococcus* spp. coagulase-negative	3.4 × 10^3^	Negative
14	*Pasteurella* spp./*Corynebacterium* spp.	3.2 × 10^3^	*Corynebacterium* spp.	5.8 × 10^3^	Negative
15	*Staphylococcus* spp. coagulase-positive	6.1 × 10^2^	*Staphylococcus* spp. coagulase-negative	1.75 × 10^3^	Negative
16	*Bacillus* spp.	1.2 × 10^2^	*Bacillus* spp.	9.2 × 10^2^	Negative
17	*Pasteurella* spp./*Corynebacterium* spp.	6.6 × 10^3^	*Pasteurella* spp./*Corynebacterium* spp.	2.9 × 10^3^	Positive
18	*Pasteurella* spp./*Corynebacterium* spp.	2.1 × 10^3^	*Enterobacter aerogenes*/*Corynebacterium* spp.	4.2 × 10^3^	Negative
19	*Ochrobacterium anthropi*/*Corynebacterium* spp.	>10^5^	*Corynebacterium* spp.	>10^5^	Negative
20	*Bacillus* spp.	1.2 × 10^3^	*Corynebacterium* spp./*Staphylococcus* spp. coagulase -positive	1.9 × 10^4^	Negative
21	*Bacillus* spp.	2.03 × 10^4^	Negative	1.2 × 10^2^	Negative
22	Negative	<100	Negative	1.3 × 10^3^	Negative
23	Negative	1.1 × 10^2^	*Bacillus* spp.	1.8 × 10^3^	Negative
24	Negative	<100	*Corynebacterium* spp.	2.3 × 10^2^	Negative
25	*Ochrobacterium anthropi*/*Bacillus* spp.	1.5 × 10^2^	*Bacillus* spp.	4.6 × 10^3^	Positive
26	*Bacillus* spp.	4.7 × 10^2^	Negative	3.4 × 10^2^	Negative
27	Negative	1.2 × 10^5^	Negative	1.1 × 10^3^	Negative

^a^ oculus dexter; ^b^ oculus sinister; ^c^ total bacteria count.

**Table 5 animals-11-02987-t005:** The normal values of tear production reported in various avian species.

Species	Tear Production	Method Used	References
Local breed chicken (*Gallus gallus domesticus*) The Livorno breed (276–375 days) The Siciliana breed (276–375 days)	21.98 ± 3.46 mm/15 s 23.77 ± 2.99 mm/15 s 19.95 ± 2.81 mm/15 s	PRTT ^6^ PRTT ^6^ PRTT ^6^	Present study
Broiler chicks (*Gallus gallus domesticus*) 42-day-old chicks 5-day-old chicks 42-day-old chicks 5-day-old chicks 42-day-old chicks 5-day-old chicks 42-day-old chicks	11.40 ± 2.60 mm/min 5.00 ± 1.83 mm/min 10.45 ± 2.58 mm/min 12.37 ± 1.80 mm/15 s 25.58 ± 4.8 mm/15 s 7.13 ± 0.72 mm/min 12.03 ± 0.92 mm/min	STTI ^3^ mSTTI ^4^ mSTTI ^4^ PRTT ^6^ PRTT ^6^ EAPPTT ^8^ EAPPTT ^8^	[29]
American flamingo (*Phoenicopterus ruber ruber*)	12.3 ± 4.5 mm/min 24.2 ± 4.4 mm/15 s	mSTTI ^4^ PRTT ^6^	[22]
American white pelicans (*Pelecanus erythrorhynchos*)	14.9 ± 7.84 mm/15 s	PRTT ^6^	[44]
Atlantic puffins (*Fratercula arctica*)	OD ^1^: 7.5 mm/15 s (IQR ^13^: 6.5–9.3) OS ^2^: 5.0 mm/15 s. (IQR ^13^: 4.0–7.3)	PRTT ^6^ PRTT ^6^	[47]
Bald eagle (*Haliaetus leucocephalus*)	14 ± 2 mm/min	STTI ^3^	[20]
Brown pelicans (*Pelecanus occidentalis*)	5.45 ± 1.88 mm/min	STTI ^3^	[41]
Cinereous vultures (*Aegypius monachus*)	OD ^1^: 11.4 ± 2.6 mm/min OS ^2^: 11.5 ± 2.8 mm/min OD ^1^: 22.3 ± 2.1 mm/15 s OS ^2^: 22.8 ± 3.0 mm/15 s	STTI ^3^ STTI ^3^ PRTT ^6^ PRTT ^6^	[46]
Common buzzard (*Buteo buteo*)	12.47 ± 2.66 mm/min	STTI ^3^	[19]
Common kestrel (*Falco tinnunculus*)	7.4 ± 3.27 mm/min	STTI ^3^	[43]
Common murres (*Uria aalge*)	20 ± 3.6 mm/15 s	PRTT ^6^	[42]
Common mynah (*Acridotheres tristis*)	LE ^11^: 19.2 ± 2.5 mm/15 s UE ^12^: 17.5 ± 3.1 mm/15 s	PRTT ^6^ PRTT ^6^	[36]
Ducks (*Anas platyrynchos*)	6.2 ± 2.2 mm/min	STTI ^3^	[25]
Eastern screech owl (*Megascops asio*)	≤2 mm/min 15 ± 4.3 mm/15 s	STTI ^3^ PRTT ^6^	[8]
Eurasian black vulture (*Aegypius monachus*)	OD ^1^: 10.9 ± 3.3 mm/min OS ^2^: 11.9 ± 3.3 mm/min	STTI ^3^ STTI ^3^	[37]
Eurasian Tawny owl (*Strix aluco*)	3.12 ± 1.92 mm/min	STTI ^3^	[19]
European kestrel (*Falco tinnunculus*)	6.20 ± 3.67 mm/min	STTI ^3^	[19]
Falconiformes (genus *Falco*)	30.6 ± 4.2 mm/15 s	PRTT ^6^	[40]
Geese (*Anser anser*)	5.5 ± 2.6 mm/min	STTI ^3^	[25]
Great grey owls (*Strix nebulosa*)	9.8 ± 2.8 mm/min	STTI ^3^	[23]
Griffon vulture (*Gyps fulvus*)	OD ^1^: 6.4 ± 1.8 mm/min OS ^2^: 6.5 ± 1.8 mm/min	STTI ^3^ STTI ^3^	[37]
Helmeted Guinea Fowl (*Numida meleagris*)	16.5 ± 1.3 mm/15 s	PRTT ^6^	[39]
Hispaniolan parrots (*Amazona ventralis*)	12.5 ± 5.0 mm/15 s 12.6 ± 5.4 mm/15 s 7.9 ± 2.6 mm/min 5.1 ± 3.3 mm/min	PRTT ^6^ (WTA ^9^) PRTT ^6^ (TA ^10^) STTI ^3^ (WTA ^9^) STTI ^3^ (TA ^10^)	[9]
Humboldt penguin (*Spheniscus humboldti*)	6.45 ± 2.9 mm/min	STTI ^3^	[10]
Humboldt penguin (*Spheniscus humboldti*)	9 ± 4 mm/min	STTI ^3^	[26]
Large Psittaciformes (various genus, species)	OD ^1^: 19.8 ± 4.3 mm/15 s OS ^2^: 20.1 ± 3.9 mm/15 s	PRTT ^6^ PRTT ^6^	[35]
Little owl (*Athene noctua*)	3.5 ± 1.96 mm/min	STTI ^3^	[19]
Macaroni penguin (*Eudyptes chrysolophus*)	24.7 ± 6.37 mm/15 s 12.1 ± 5.43 mm/min	mPRTT ^7^ STTII ^5^	[38]
Nocturnal raptor of different species Diurnal raptor of different species	16.5 ± 7.6 mm/15 s 3.4 ± 3.8 mm/min 23.5 ± 8.9 mm/15 s 8.3 ± 5.4 mm/min	PRTT ^6^ STTI ^3^ PRTT ^6^ STTI ^3^	[34]
Ostrich (*Struthio camelus*)	16.3 ± 2.5 mm/min	STTI ^3^	[18]
Pigeons (*Turkish pigeons—Ankut trumpeter*)	April: 23.02 ± 2.98 mm/15 s June: 24.04 ± 2.60 mm/15 s	PRTT ^6^ PRTT ^6^	[21]
Snowy owls (*Bubo scandiacus*)	9.8 ± 2.4 mm/min	STTI ^3^	[23]
Southern rockhopper penguin (*Eudyptes chrysocome*)	25.1 ± 7.07 mm/15 s 11.0 ± 3.96 mm/min	mPRTT ^7^ STTII ^5^	[38]
Whooper swans (*Cygnus cygnus*)	22.59 ± 3.48 mm/15 s	PRTT ^6^	[45]

^1^ oculus dexter; ^2^ oculus sinister; ^3^ Schirmer’s tear test 1; ^4^ modified Schirmer’s tear test 1; ^5^ Schirmer’s tear test 2; ^6^ phenol red thread test; ^7^ modified phenol red thread test; ^8^ endodontic absorbent paper points tear test; ^9^ without topical anesthesia; ^10^ topical anesthesia; ^11^ lower eyelids; ^12^ upper eyelid; ^13^ interquartile range.

**Table 6 animals-11-02987-t006:** Normal values for intraocular pressure (IOP) reported in various avian species.

Species	Mean ± SD IOP ^1^ (mmHg)	Device Calibration	References
Local breed chicken (*Gallus gallus domesticus*) The Livorno breed The Siciliana breed	14.19 ± 1.16 14.3 ± 1.17 14.06 ± 1.15	TV-P ^5^	Present study
White Leghorn chickens (*Gallus gallus domesticus*)	Day: 20.8 ± 1.1 Night: 16.8 ± 0.7	TP ^4^ TP ^4^	[33]
White Leghorn chickens (*Gallus gallus domesticus*)	Day: 22.3 Night: 12.3	TP ^4^ TP ^4^	[1]
White Leghorn/Broiler intercross (*Gallus gallus domesticus*) White Leghorn (*Gallus gallus domesticus*) Broiler (*Gallus gallus domesticus*)	17.51 ± 0.13 16.3 ± 0.2 16.1 ± 0.2	TV-P ^5^ TV-P ^5^ TV-P ^5^	[7]
American flamingo (*Phoenicopterus ruber ruber*)	16.1 ± 4.2 9.5 ± 1.7	TP ^4^ TV-P ^5^	[22]
American flamingo (*Phoenicopterus ruber*)	OS ^3^: 11.1 ± 2.3 OD ^2^: 10.9 ± 1.8	TV-P ^5^	[52]
American kestrel (*Falco sparverius*)	8.5 ± 4.4 6.8 ± 1.7	TP ^4^ TV-P ^5^	[16]
American white pelicans (*Pelecanus erythrorhynchos*)	9.0 ± 1.41	TV-P ^5^	[44]
Atlantic puffins (*Fratercula arctica*)	13 (range 12–15)	TV-P ^5^	[47]
Bald eagle (*Haliaetus leucocephalus*)	21.5 ± 1.7	TP ^4^	[20]
Barn owl (*Tyto alba*)	10.8 ± 3.8	TV-D ^6^	[15]
Barred owl (*Strix varia*)	11.7 ± 3.8 8.3 ± 3.2	TP ^4^ TV-P ^5^	[16]
Black-footed penguin (*Spheniscus demersus*)	OD ^2^: 30.41 ± 4.27 OS ^3^: 28.13 ± 6.84 OD ^2^: 25.06 ± 4.35 OS ^3^: 25.05 ± 5.56	TV-D ^6^ TV-H ^7^	[11]
Brown pelicans (*Pelecanus occidentalis*)	10.86 ± 1.61	TP ^4^	[41]
Cinereous vultures (*Aegypius monachus*)	OD ^2^: 32.8 ± 6.9 OS ^3^: 31.9 ± 7.1 OD ^2^: 20.7 ± 4.5 OS ^3^: 19.5 ± 4.1	TV-D ^6^ TV-D ^6^ TP ^4^ TP ^4^	[46]
Common buzzard (*Buteo buteo*)	26.9 ± 7.0	TV-D ^6^	[15]
Common buzzard (*Buteo buteo*)	17.2 ± 3.53	TP ^4^	[19]
Common kestrel (*Falco tinnunculus*)	10.5 ± 3.15	TP ^4^	[43]
Common murres (*Uria aalge*)	22.7 ± 2.6	TV-D ^6^	[42]
Cooper’s hawk (*Accipiter cooperii*)	16.0 ± 1.8 10.7 ± 1.4	TP ^4^ TV-P ^5^	[16]
Domestic pigeon (*Columbia livia)*	6.1 ± 0.9	TV-P ^5^	[24]
Ducks (*Anas platyrynchos*)	10.2 ± 2.2	TV-P ^5^	[25]
Eastern screech owl (*Megascops asio*)	11.0 ± 1.9 9.0 ± 1.8 14.0 ± 2.4	TP ^4^ TV-P ^5^ TV-D ^6^	[8]
Eastern screech owl (*Megascops asio*)	9.3 ± 2.6 6.3 ± 1.3	TP ^4^ TV-P ^5^	[16]
Eurasian eagle owl (*Bubo bubo*)	9.35 ± 1.81 10.45 ± 1.64	TP ^4^ TV-P ^5^	[6]
Eurasian Tawny owl (*Strix aluco*)	11.21 ± 3.12	TP ^4^	[19]
European kestrel (*Falco tinnunculus*)	9.8 ± 2.5	TV-D ^6^	[15]
European kestrel (*Falco tinnunculus*)	8.53 ± 1.59	TP ^4^	[19]
Geese (*Anser anser*)	9.1 ± 2.0	TV-P ^5^	[25]
Great grey owls (*Strix nebulosa*)	9.6 ± 2.6	TV-P ^5^	[23]
Great horned owl (*Bubo virginianus*)	9.9 ± 2.4 9.9 ± 2.2	TP ^4^ TV-P ^5^	[16]
Helmeted Guinea Fowl (*Numida meleagris*)	9.1 ± 0.9	TV-P ^5^	[39]
Humboldt penguin (*Spheniscus humboldti*)	20.36 ± 4.1	TP ^4^	[9]
Humboldt penguin (*Spheniscus humboldti*)	28 ± 9	TV-P ^5^	[26]
Little owl (*Athene noctua*)	9.83 ± 3.41	TP ^4^	[19]
Long-eared owl (*Asio otus*)	7.8 ± 3.2	TV-D ^6^	[15]
Macaroni penguin (*Eudyptes chrysolophus*)	21.9 ± 7.05 29.1 ± 7.16	TP ^4^ TV-D ^6^	[38]
Nocturnal raptor of different species Diurnal raptor of different species	15.4 ± 5.7 14.9 ± 4.8	TP ^4^ TP ^4^	[34]
Northern goshawk (*Accipiter gentilis*)	18.3 ± 3.8	TV-D ^6^	[15]
Ostrich (*Struthio camelus*)	18.8 ± 3.5	TP ^4^	[18]
Peregrine falcon (*Falco peregrinus*)	12.7 ± 5.8	TV-D ^6^	[15]
Red kite (*Milvus milvus*)	13.0 ± 5.5	TV-D ^6^	[15]
Red-tailed hawk (*Buteo jamaicensis*)	20.3 ± 2.8 19.8 ± 4.9	TP ^4^ TV-P ^5^	[16]
Snowy owls (*Bubo scandiacus*)	9.1 ± 1.9	TV-P ^5^	[23]
Southern rockhopper penguin (*Eudyptes chrysocome*)	20.0 ± 5.77 24.1 ± 5.09	TP ^4^ TV-D ^6^	[38]
Sparrow hawk (*Accipiter nisus*)	15.5 ± 2.5	TV-D ^6^	[15]
Tawny owl (*Strix aluco*)	15.6 ± 3.4	TP ^4^	[4]
Tawny owl (*Strix aluco*)	9.4 ± 4.1	TV-D ^6^	[15]
Turkey vulture (*Cathartes aura*)	15.0 ± 2.1 11.7 ± 1.0	TP ^4^ TV-P ^5^	[16]
White-tailed sea eagle (*Haliaeetus albicilla*)	26..9 ± 5.8	TV-D ^6^	[15]
Whooper swans (*Cygnus cygnus*)	11.30 ± 3.55	TV-D ^6^	[45]

^1^ intraocular pressure; ^2^ oculus dexter; ^3^ oculus sinister; ^4^ Tonopen; ^5^ TonoVet with no specific species calibration; ^6^ TonoVet with dog calibration setting; ^7^ TonoVet with horse calibration setting.

## Data Availability

The data presented in this study are available on request from the corresponding author.

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
