# Peer review of "Tear Production, Intraocular Pressure, Ultrasound Biometric Features and Conjunctival Flora Identification in Clinically Normal Eyes of Two Italian Breeds of Chicken (Gallus gallus domesticus)"

_animals, 2021, doi:10.3390/ani11102987_

Round 1
Reviewer 1 Report
The study shows the results for a series of ocular parameters measured in chickens of 2 Italian breeds. The manuscript provides an adequate description of how the study was carried out, and the authors provide a large number of data from other papers, obtained in the same and other species, for comparison.
After having reviewed it, my recommendations to improve the manuscript would be the following:
- Please, revise the scientific names of the species throughout the text and tables, both for birds and microorganisms, as they should be in italics, including the section “References”.
- Unify the expression of units: in the text the symbol "sec" for second appears several times and in the tables second is represented as "s" (Table 1). The international measurement system indicates that the symbol for second, regardless of the language, is "s". Separate also the numbers of the corresponding units throughout the text.
- In tables 1, 2 and 3, tables a, b and c included in each of them should be joined so that the data for the whole population and those obtained for each breed are summarized in the same table. Why is "n" not always 31 and 35 for each breed? It is not explained anywhere why certain measurements are made with a smaller number of animals.
- The results in table 4 should not be presented for each individual but as a whole, aggregated by microorganisms/genus, indicating the number of isolates obtained for each of the bacterial species. This would give a better idea of the frequency of isolation for each microorganism. It could also include the range for the TBC.
- Tables 5 and 6 include a large number of data for comparison. However, the results of the study itself and those of other studies carried out on the same animal species should appear first, followed by those of the birds phylogenetically closest to the chicken (then the rest). Another option would be for all species, outside the chicken, to be presented alphabetically. And the same order and criteria should be followed, as far as possible, in both tables.
- It is not correct to say "statistically significant" but “significant".
- References should be checked, as sometimes "&" appears before the last author.
- Lines 42-43. The paragraph seems to have been split into two.
Reviewer 2 Report
Interesting study and of great practical importance. The presented results may be of use to veterinarians and poultry farmers.
I am of an opinion that the article fits into scope of Animals and could be published after some minor corrections.
1. Please add the scientific name of chicken to the keywords.
2. Introduction: Please add full name (author and date when species was published and described) of the chicken: chicken Gallus gallus domesticus (Linnaeus, 1758).
3. Scientific names (genus, species) should be italicized. Species (second) name should be in lower case (Table 5, Pelecanus erythrorhynchos).
4. After „spp” should be dot.
5. Table 4: What does the symbol "*" mean.
6. Table 5 and 6: please correct the title (first column). “Eastern screech owl” etc. is a common name, not breed; it will be enough “Species/subspecies”, Gallus gallus domesticus is a subspecies. I am not sure if "large" is the correct, valid scientific name of Psittaciformes.
7. Please correct references list according to the Animals: e.g. titles (journal) should be lowercase (e.g. 17, 47); please correct abbreviations, e.g. "Vet. Ophthalmol. (dots); scientific names must be italicized.
Reviewer 3 Report
This is a work conducted with rigorous methodology and clearly explained. The study of some ocular parameters in avian species widely diffused in Italy is particularly useful to be able to recognize early signs of ocular diseases in these animals. Furthermore, these parameters are fundamental in order to carry out experimental studies in these species.
However, there are some important points that need to be clarified before proceeding with publication, and the language style in some lines is not immediately understandable.
Line 118 Can you explain why before ultrasound examination topical anaehstehesia was not performed? this is an important issue in respect of animal welfare.
Line 162: Please, specificate that you evaluated if there were significative differences of PRTTs, IOPs, D1s, D2s, D3s, D4s and D5s between right and left eyes. If there are statistical differences they can’t be considered as a single value for each bird.
Line 208. Can you add an explicative photografic imagine of ultrasound measurements in a chicken?
Line 319. Please specify if your data and those of Prasher were both reported as mean values. In addition, there were weight differences between the two populations of chicken tested in the two different studies?
Line 346-352 : please be aware that these periods are repetitions
Round 2
Reviewer 1 Report
Please, include a sentence indicating that not all animals tolerated well procedures and some were excluded (to explain why n is not always the same).
Author Response
We have added a sentence, as you suggested, in "data collection" (line 83-85) in the "materials and methods" section as follows:
"Ultrasound biometric measurements, and collection of samples for conjunctival flora identification were performed in 27 out of 66 chickens because the birds that exhibited excessive stress during the procedures were excluded."
We have also specified the number of chickens enrolled in the study in the paragraphs relating to ultrasound biometric measurements (line 118) and sampling for the identification of the conjunctival flora (line 130).
The changes were made in red in the main text.